# Feasibility, Reproducibility and Reference Ranges of Left Atrial Strain in Preterm and Term Neonates in the First 48 h of Life

**DOI:** 10.3390/diagnostics12020350

**Published:** 2022-01-29

**Authors:** Benjamim Ficial, Iuri Corsini, Maria Clemente, Alessia Cappelleri, Giulia Remaschi, Laura Quer, Giulia Urbani, Camilla Sandrini, Paolo Biban, Carlo Dani, Giovanni Benfari

**Affiliations:** 1Neonatal Intensive Care Unit, Azienda Ospedaliera Universitaria Integrata Verona, 37126 Verona, Italy; maria.clemente.889@gmail.com (M.C.); paolo.biban@aovr.veneto.it (P.B.); 2Division of Neonatology, Careggi University Hospital of Florence, 50134 Florence, Italy; corsiniiuri@gmail.com (I.C.); giulia.remaschi@gmail.com (G.R.); cdani@unifi.it (C.D.); 3Neonatal Intensive Care Unit, Fondazione IRCCS Ca’ Granda Ospedale Maggiore Policlinico, 20122 Milano, Italy; aly0288@gmail.com; 4Section of Cardiology, Department of Medicine, University of Verona, 37126 Verona, Italy; laura.quer@gmail.com (L.Q.); urbani.giu@gmail.com (G.U.); camilla.sandrini@aovr.veneto.it (C.S.); giovanni.benfari@gmail.com (G.B.)

**Keywords:** left atrial strain, neonate, echocardiography, speckle tracking, neonatal intensive care, preterm

## Abstract

Left atrial strain (LAS) is the most promising technique for assessment of diastolic dysfunction but few data are available in neonates. Our aim was to assess feasibility and reproducibility, and to provide reference ranges of LAS in healthy neonates in the first 48 h of life. We performed one echocardiography in 30 neonates to assess feasibility and develop a standard protocol for image acquisition and analysis. LAS reservoir (LASr), conduit (LAScd) and contraction (LASct) were measured. We performed echocardiography at 24 and 48 h of life in an unrelated cohort of 90 neonates. Median (range) gestational age and weight of the first cohort were 34.4 (26.4–40.2) weeks and 2075 (660–3680) g. LAS feasibility was 96.7%. Mean (SD) gestational age and weight of the second cohort were 34.2 (3.8) weeks and 2162 (833) g. Mean (SD) LASr significantly increased from 24 to 48 h: 32.9 (3.2) to 36.8 (4.6). Mean (SD) LAScd and LASct were stable: −20.6 (8.0) and −20.8 (9.9), −11.6 (4.9) and −13.5 (6.4). Intra and interobserver intraclass correlation coefficient for LASr, LAScd and LASct were 0.992, 0.993, 0.986 and 0.936, 0.938 and 0.871, respectively. We showed high feasibility and reproducibility of LAS in neonates and provided reference ranges.

## 1. Introduction

Echocardiographic assessment of left ventricle (LV) filling pressure is an essential step in the assessment of diastolic dysfunction [1,2]. Most widespread echocardiographic indexes used for this purpose are obtained with Doppler or Tissue Doppler techniques and may have limitations due to angle dependence [3,4].

In adult patients, two-dimensional (2D) speckle tracking echocardiography (STE) has been applied to assess left atrium (LA) function and left atrial strain (LAS) has recently been proposed as novel technique to be used in clinical practice [3]. LAS measurements can be easily obtained, are not angle dependent, strongly correlate with invasive measurement of LV filling pressure and have a prognostic role in adult diastolic heart failure, atrial fibrillation and after acute myocardial infarction [5,6].

LAS is the most promising technique for direct evaluation of LA function and non-invasive assessment of LV diastolic function in adults [7]. Different echo views for imaging acquisition and different software packages and modalities of postprocessing analysis are available. A recent consensus document has been published to standardize techniques and definitions for using 2D STE to assess LA deformation [8].

To the best of our knowledge, few data on LAS are available in neonates and no data are available in the first days of life, also known as ‘transitional period’, when the transition from intrauterine to extrauterine circulation occurs [9,10].

LA function has been conventionally divided into three phases timed to the ventricular QRS complex. During ventricular contraction, LA acts as reservoir by storing incoming pulmonary venous blood. During early ventricular relaxation, blood flows passively into the ventricle and LA acts as conduit between the pulmonary veins and LV. In the last phase, LA contracts and actively pumps blood into LV [11].

In neonates, there is evidence of an early diastolic dysfunction in the transitional period, particularly when born prematurely [12,13,14]. Moreover, important changes of LV function and LV filling pressure occur in the transitional period, likely related to changes in loading conditions, decreases in pulmonary vascular resistance and the presence of residual fetal shunts, such as patent ductus arteriosus (PDA) [15]. Therefore, there may be a role for LAS to be used in the assessment of atrial function in neonates in the transitional period.

The first aim of our study was to assess feasibility of LAS in healthy preterm and term neonates in the first 48 h of life and to develop a standard protocol for image acquisition and postprocessing analysis. The second aim was to provide reference ranges of LAS measurements in healthy preterm and term neonates in the first 48 h of life, to assess LAS reproducibility and to explore associations with other echocardiographic parameters of left systolic and diastolic function.

## 2. Methods

### 2.1. Study Design

This was a multicenter prospective observational cohort study conducted at the neonatal intensive care unit (NICU) and postnatal ward of Careggi University Hospital of Florence (Italy) and University Hospital of Verona (Italy). The study was divided into two phases.

In the first phase, we developed an LAS protocol for image acquisition and postprocessing analysis and we assessed its feasibility in a cohort of 30 healthy neonates. Every infant underwent one echocardiography in the first 48 h of life.

In the second phase, in an unrelated cohort of 90 healthy neonates, we applied this protocol to provide reference ranges of LAS at two timepoints (T_1_ = 24 ± 2 h and T_2_ = 48 ± 2 h), to assess the association of LAS measurements with other echocardiographic parameters and to assess LAS intraobserver and interobserver reproducibility. Ethics approval was obtained from research ethics committees of University Hospital of Florence (Italy) and University Hospital of Verona (49/2016 and 2132CESC respectively). Parents or guardians of babies were approached in the NICU or postnatal ward and gave their written informed consent prior to enrollment.

### 2.2. Study Population

In the first phase, from December 2018 to March 2019, we prospectively recruited 30 healthy neonates, divided according to gestational age (GA) as follows: 10 term neonates with GA ≥ 37 weeks; 10 preterm neonates with GA between 32 and 37; and 10 preterm neonates with GA ≤ 32 weeks.

In the second phase, from April 2019 to April 2020, we prospectively recruited 90 healthy neonates divided as follows: 30 term neonates with GA ≥ 37 weeks; 30 preterm neonates with GA between 32 and 37; and 30 preterm neonates with GA ≤ 32 weeks.

Inclusion criteria were informed parental consent and cardio-respiratory healthiness. The latter was defined as follows: no need for supplemental oxygen and/or respiratory support, except for babies with GA ≤ 32 weeks, in whom non-invasive respiratory support with a FiO_2_ < 0.3 was accepted, as previously reported [16,17]. Exclusion criteria were suspected intrauterine infection, suspected sepsis, major congenital abnormalities, congenital heart disease, except for PDA and patent foramen ovale (PFO), small or large for gestational age, intrauterine growth retardation, pulmonary hypertension, need for inotropes and/or vasopressors or infants of diabetic mothers.

### 2.3. Echocardiography

All echocardiography examinations were performed using a Philips Epiq 7 system with 12 MHz probe (Philips Ultrasound, Andover, MA, USA). Images were acquired according to the recommendations of the American Society of Echocardiography [18]. Electrocardiograms were recorded simultaneously, and three cardiac cycles were digitally stored in high frame rate (100–130 Hz). Digitally stored images of LA were reviewed, and image quality was independently assessed according to Colan et al. [19]. The system has five grade levels: excellent, good, fair, poor and unusable. Images rated poor or unusable were excluded. Included images were analyzed offline using Tomtec Arena, version TTA2 41.00, with dedicated LV and LA analysis option (Tomtec, Unterschleißheim, Germany). As recommended in adults, we selected the QRS onset, surrogate of end diastole, as the reference point for starting LAS analysis [8]. The following measurements were automatically calculated from the strain curve (Figure 1):-LASr = strain during reservoir phase, calculated as the difference of the strain value at mitral valve opening minus ventricular end-diastole (positive value);-LAScd = strain during conduit phase, calculated as the difference of the strain value at the onset of atrial contraction minus mitral valve opening (negative value);-LASct = strain during contraction phase, calculated as the difference of the strain value at end-diastole minus onset of atrial contraction (negative value) [8,20].

### 2.4. First Phase

Clinical characteristics of the enrolled neonates were collected at the time of scanning. We evaluated possible options in image acquisitions and postprocessing. Adult recommendations suggest assessing monoplane LAS from a single apical 4-chamber view but computation of biplane LAS, from both apical 4- and 2-chamber views, may be an option. We assessed feasibility of the monoplane LAS and biplane LAS as well as their agreement. A standard protocol for LAS was developed.

### 2.5. Second Phase

Clinical characteristics of the enrolled neonates were collected at the time of scanning at the two timepoints (T_1_ and T_2_). LAS measurements were carried out according to the protocol developed in the first phase. LV global longitudinal strain (GLS) and strain rate (SR) were calculated as previously reported [21]. The following echocardiographic measurements were carried out as previously described: shortening fraction (SF), ejection fraction (EF) [18,22], mitral valve Doppler E wave, A wave and E/A ratio, Tissue Doppler Imaging e’, a’, s’; E/e’ ratio [23], left ventricular output [24], PDA diameter, PDA maximum velocity, left pulmonary artery (LPA) diastolic velocity and reverse diastolic flow in descending aorta [25]. A PDA was defined as hemodynamically significant (hsPDA) if the following conditions were fulfilled at 48 h: (1) PDA diameter was > 1.5 mm; (2) maximum velocity of flow across PDA was < 2 m/s; (3) LPA diastolic velocity was > 0.2 m/s; (4) LVO was > 200 mL/kg/min and (5) there was evidence of reverse diastolic flow in descending aorta [25,26].

Intraobserver and interobserver reproducibility were assessed in a random subset of 23 neonates. LAS measurements were repeated by the primary observer 4 weeks after the first measurements and by a second observer blinded to previous data, as reported [27,28].

### 2.6. Statistical Analysis

Data are expressed as mean and standard deviation (SD) for parametric variables, median (range) for non-parametric variables and percentage for categorical variables. Reference ranges of LAS measurements were reported as mean ± 2 SD. Normality was assessed using the Kolmogorov–Smirnov test. A paired-samples *t*-test was used to assess differences between echocardiographic parameters at the two timepoints. To assess associations between LAS measurements and other echocardiographic parameters of LV systolic and diastolic function, Pearson’s correlation coefficients were calculated. The sample size for reproducibility assessment was calculated considering a minimum acceptable reliability of 0.7, number of raters per subject of 2 and assuming an expected reliability of 0.9. To reach an 80% power at 0.05 level, the sample size was 23 scans [29]. Agreement, intra and interobserver reproducibility were assessed using Bland–Altman analysis (mean bias and limits of agreement, LOA) and intraclass correlation coefficient (ICC) with 95% confidence intervals (CI); *p* values < 0.05 were considered significant. Data were analyzed with SPSS 20 (SPSS, Chicago, IL, USA).

## 3. Results

### 3.1. First Phase

Median (range) gestational age and weight of the 30 neonates recruited in the first phase of the study were 34.4 (26.4–40.2) weeks and 2075 (660–3680) g respectively. Clinical characteristics of the three sub-cohorts of neonates divided according to GA are detailed in Table 1.

Monoplane LAS from a single apical 4-chamber view and biplane LAS from both a 4- and 2-chamber view were calculated. Quality of 29 out of 30 apical 4-chamber view images was rated as excellent or good, with a resulting feasibility of 96.7%, whereas only 26 out of 30 apical 2-chamber view images were rated excellent or good with a resulting feasibility of 86.7% (Appendix A).

There was excellent agreement and correlation between monoplane and biplane LAS measurements (Table 2). We developed a protocol for LAS imaging and postprocessing data analysis (see Appendix B).

### 3.2. Second Phase

Mean (SD) gestational age and weight of the 90 neonates recruited in the second phase of the study were 34.2 (3.8) weeks and 2162 (833) g respectively. Clinical characteristics of the three sub-cohorts of neonates divided according to GA are shown in Table 3. We did not find statistically significant differences in heart rate between T_1_ and T_2_ in the three groups of neonates (*p* > 0.05).

Quality of 171 out of 180 apical 4-chamber view images was rated excellent or good with a resulting feasibility of 95% (Appendix A).

We observed excellent intraobserver reproducibility and slightly inferior interobserver reproducibility (Table 4).

In all 90 neonates, mean (SD) LASr significantly increased from T_1_ and T_2_: 32.9 (3.2) to 36.8 (4.6). However, mean (SD) LAScd and LASct did not show statistically significant differences between the two timepoints: −20.6 (8.0) and−20.8 (9.9) and −11.6 (4.9) and−13.5 (6.4) respectively. 

Reference ranges (mean ± 2SD) of LASr, LAScd and LASct in the three groups of neonates divided according to GA are shown in Table 5.

In each of the three groups of neonates LASr significantly increased from T_1_ to T_2_. LAScd did not show statistically significant differences between T_1_ and T_2_ in all groups, whereas we found a statistically significant difference between LASct at T_1_ and T_2_ only in neonates with GA between 32 and 37 weeks.

Echocardiographic parameters of left systolic function, left diastolic function and LVO at T_1_ and T_2_ are presented in Table 6.

There was a weak negative linear correlation between LASr and both GLS and E/e’ at T_1_ and T_2_ and between LAScd and E/e’ at T_1._ (Table 7).

Finally, we compared LAS measurements in preterm neonates (GA ≤ 32 weeks) with and without hsPDA, defined as mentioned above (see Methods section). Nine neonates (30%) had hsPDA. LASr and LASct were significantly different in neonates with and without hsPDA, whereas LAScd did not show any statistical significance. Other variables of systolic (GLS) and diastolic function (E, A, E/A and E/e’) did not show statistically significant differences (Table 8).

## 4. Discussion

To the best of our knowledge, this is the first study that longitudinally assessed LAS measured by 2D STE in term and preterm neonates in the first 48 h of life, during the transition from intrauterine to extrauterine circulation. We standardized image acquisition and postprocessing analysis and demonstrated high feasibility and reproducibility of LAS measurements. Finally, we provided reference ranges in healthy preterm and term neonates during the transitional period, that can be used as reference data for future research on diastolic dysfunction in neonates with heart failure and/or volume overload.

A major challenge to widespread clinical application of LAS is the lack of standardization of image acquisition and postprocessing analysis, different software packages and parameters to be measured. A recent consensus document tried to standardize definitions and techniques for using LAS in adults. The same would be needed in neonates, given the fact that the current available data were obtained with different types of software and different modalities of both image acquisition and postprocessing analysis [8,9,10,30].

Firstly, it is essential to use software with a dedicated LA option. Cantinotti et al. presented data on LAS measurement in children using software with a left ventricular option, but they excluded the neonatal subgroup because the feasibility was very low in neonates [30].

On the contrary, we showed that LAS had an optimal feasibility, as previously reported in neonates outside the transitional period and in children, when software with a dedicated LA option was used [9,10].

In the first phase of the study, we developed an LAS protocol for image acquisition and postprocessing analysis. The recent consensus statement recommended that LAS should be computed using a monoplane algorithm from an apical, non-foreshortened, 4-chamber view or, if available, from a biplane algorithm, which includes the apical 4-chamber and 2-chamber views, whereas assessment of LAS from an apical 3-chamber view should be avoided because the ascending aorta is difficult to separate from the atrial wall [8]. De Waal et al. computed LAS using a triplane algorithm, that included LAS from an apical 3-chamber view, although they acknowledged that image acquisition needed further standardization, given the complexity of having good quality images. Finally, they concluded that LA monoplane analysis might improve reliability and standardization [10].

In our cohort of neonates, we found excellent agreement between biplane and monoplane LAS, and from a practical perspective, they seem to be interchangeable. Due to the fact that feasibility of the apical 4-chamber view was superior compared to the apical 2-chamber view, we chose to include only a calculation of monoplane LAS from an apical 4-chamber view in our protocol.

In conclusion, in our opinion, computing LAS only from an apical 4-chamber view is preferrable due to its superior feasibility, accuracy and time-efficiency.

We demonstrated excellent intraobserver reproducibility for LASr, LAScd and LASct and slightly inferior interobserver reproducibility. We feel that this was achieved by developing a standard protocol for image acquisition and postprocessing analysis. Our data on reproducibility are similar to that reported by De Waal et al. and Kutty at al. and confirm that within and between observer differences in LAS measurements may not be clinically relevant [9,10].

Although LAS is currently the best available non-invasive technique to directly assess LA function and indirectly assess LV diastolic function in adults, few data on LAS are available in neonates, particularly if born prematurely. Two pediatric studies presented normal values of LAS in term neonates and older children. Kutty et al. and Ghelani et al. reported reference values of LAS in subjects with an age ranging from 3 days to 20 years and from 4 days to 20.9 years, respectively [9,31]. However, there is only one study by De Waal et al. that reported data on LAS in preterm neonates with GA < 30 weeks at 3 days and four weeks of postnatal age [10]. These are the first data on LAS in both term and preterm neonates at 24 and 48 h of life.

We reported reference values for the components of LAS in the first 48 h of life in a cohort of preterm and term neonates. We showed a significant increase in LASr in the first 48 h of life. During the transitional period, lung aeration and distension lead to a dramatic reduction in pulmonary vascular resistance and a subsequent increase in pulmonary blood flow and LV preload [15,32]. We hypothesized that the increase in atrial reservoir function reflects the above-mentioned increase in pulmonary blood flow. To support our findings, Kutty et al. showed an overall increase in LA reservoir function during the first two years of life, likely related to maturational changes of LA [9].

In our cohort of neonates, we confirmed that atrial and ventricular function are closely interdependent. Although the association is weak, GLS was the major determinant of LASr. In fact, longitudinal shortening of LV contributes to atrial stretching during systole and subsequent atrial longitudinal strain (LASr) [33]. Enlargement of LA, that may be caused by reduced LV function, leads to decreased atrial reservoir longitudinal strain for a given reservoir volume [34]. The latter is confirmed by data from children with functionally single ventricle heart disease. In this category of patients, the single atrium is dilated and has reduced reservoir function, with increased reliance on atrial contraction for ventricular filling. LAS in single ventricle patients deviates significantly from normal controls and appears similar to patients with early LV diastolic dysfunction [35].

Moreover, LV diastolic function or compliance may influence LV filling pressure and therefore both atrial reservoir and conduit functions. This is confirmed by the associations we found between E/e’, LASr and LAScd [34]. We could not evaluate how LAS measurement changes in diastolic and systolic dysfunction, because none of the neonates had overt heart failure. Nevertheless, the latter was beyond the aims of our study, where we provided reference ranges of LAS measurements in a cohort of healthy neonates. Further studies are needed to explore the clinical role of LAS measurements in this population.

Finally, our data on preterm neonates with hsPDA confirmed the findings by De Waal et al. [10]. In neonates with hsPDA, the volume overload caused by the left to right shunt through the PDA leaded to reduced atrial function compared to controls. However, echocardiographic parameters of systolic and diastolic ventricular function did not differ between cases and controls. Therefore, LAS may be a promising tool for early identification of neonates with volume overload due to hsPDA. Further studies are needed to investigate the role of LAS in the assessment of the hemodynamically significance of PDA in preterm neonates.

### Limitations

Our study has several limitations. First, the number of subjects in the three group of neonates is relatively small to establish normal ranges. Further studies are needed to confirm our findings. 

Secondly, the main limitation of 2D STE can be partial displacement of the speckles outside the imaging plane. Data on LAS measured by 3D echocardiography were recently reported in children, but no preterm neonates were included [31].

Thirdly, we did not explore differences in LAS measurements related to the two different reference points for strain curve: QRS and P wave. According to data in adults, using QRS as reference point leads to slightly superior feasibility and is less time consuming [20]. Moreover, it allows for an easier assessment of LA reservoir function, the LAS parameter that is more clinically useful in adults and children [20]. Nevertheless, data comparing the two methods in neonates are lacking.

Fourthly, we did not assess scan–rescan echo repeatability. However, a number of other studies on neonatal echocardiography presented reproducibility data only from off-line analyses [20,28,36].

Lastly, we did not investigate any association between LAS measurements and clinically relevant outcomes, because the latter is outside the scope of the current study. Further studies are required to explore the role of LAS in neonates with heart failure and volume overload.

## 5. Conclusions

This study demonstrates the high feasibility of LAS measurements by 2D STE in preterm and term neonates in the first 48 h of life and provides a protocol for image acquisition and postprocessing analysis that reduces measurement variability to a clinically insignificant level. In addition, reference ranges of LAS measurements have been presented. Further studies are required to confirm our findings and to explore the clinical utility of LAS in this population prior to implementation of LAS in clinical practice.

## Figures and Tables

**Figure 1 diagnostics-12-00350-f001:**
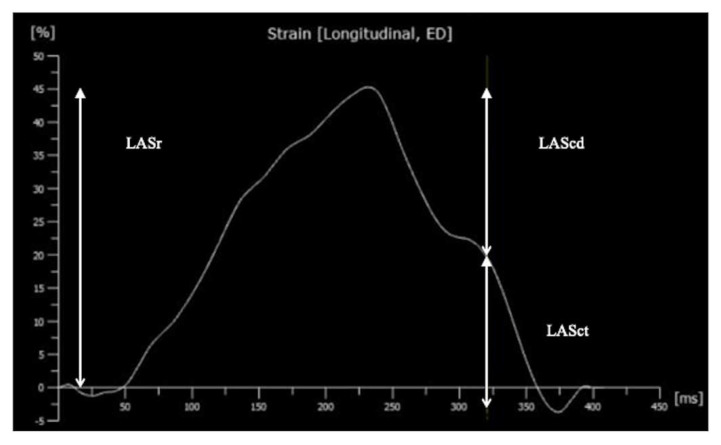
Longitudinal atrial deformation curve during the three phases of the left atrial with the zero strain reference at end-diastole. The respective strains are: LASr, strain during reservoir phase; LAScd, strain during conduit phase; and LASct, strain during contraction phase.

**Table 1 diagnostics-12-00350-t001:** Clinical characteristics of the first cohort of neonates.

N = 30	GA ≤ 32 Weeks (10 Neonates)	32 < GA < 37 Weeks (10 Neonates)	GA ≥ 37 Weeks (10 Neonates)
Gestation (weeks)	30.9 (26.4–32.0)	34.4 (33.8–36.1)	39.1 (38.1–40.2)
Birth weight (g)	1270 (660–1450)	2075 (1840–2850)	3230 (2800–3680)
Female	6 (60%)	4 (40%)	4 (40%)
Cesarean delivery	8 (80%)	7 (70%)	3 (30%)
5-Min Apgar score	8 (7–9)	9 (8–10)	10 (9–10)
Postnatal age at scan (h)	27 (24–48)	33 (24–48)	29 (24–48)
Non-invasive respiratory support with FiO_2_ < 0.3	3 (30%)	0 (0%)	0 (0%)
No respiratory support	7 (70%)	10 (100%)	10 (100%)

Values are presented as median (range), and count (%). GA = Gestational age.

**Table 2 diagnostics-12-00350-t002:** Comparison between monoplane LAS measurements from a single apical 4-chamber view and biplane LAS measurements form an apical 4- and 2-chamber view.

Comparison between Monoplane and Biplane LAS Measurements:	Bias (95% LOA)	ICC (95% CI)
LASr	−0.3 (−3.6 to 3)	0.967 (0.929 to 0.985)
LAScd	−0.4 (−3.4 to 2.6)	0.959 (0.911 to 0.982)
LASct	−0.5 (−3.5 to 3)	0.937 (0.866 to 0.971)

All ICC *p* values < 0.01. Abbreviations: LAS = left atrial strain; LASr = left atrial strain reservoir; LAScd = left atrial strain conduit; LASct = left atrial strain contraction; CI = confidence intervals; ICC = intraclass correlation coefficient; LOA = limits of agreement.

**Table 3 diagnostics-12-00350-t003:** Clinical characteristics of the second cohort of neonates.

N = 90	GA ≤ 32 Weeks (30 Neonates)	32 < GA < 37 Weeks (30 Neonates)	GA ≥ 37 Weeks (30 Neonates)
Gestational age (week)	30.4 (1.6)	34.4 (1.2)	39.6 (1.0)
Birth weight (g)	1385 (301)	2148 (304)	3348 (414)
Female	15 (50%)	13 (43.3%)	17 (56.6%)
Cesarean delivery	28 (93.3%)	27 (90%)	10 (33.3%)
5-min Apgar score	9 (7–10)	9 (8–10)	10 (8–10)
	**First Echo at 24 h of Life**		
Postnatal age (h)	24.0 (0.3)	24.1 (0.3)	24.0 (0.4)
Weight (g)	1334 (286)	2068 (301)	3224 (412)
Parenteral hydration (mL/kg/day)	79 (18)	56 (20)	NA
Heart rate (bpm)	145 (24)	132 (14)	125 (17)
Systolic blood pressure (mmHg)	64 (10)	68 (6)	74 (5)
Mean blood pressure (mmHg)	44 (8)	45 (5)	47 (4)
Diastolic blood pressure (mmHg)	36 (9)	36 (8)	34 (3)
Non-invasive respiratory support	23 (76.7%)	0 (0%)	0 (0%)
No respiratory support	7 (23.3%)	30 (100%)	30 (100%)
	**Second Echo at 48 h of Life**		
Postnatal age (h)	48.1 (0.4)	48.2 (0.5)	48.4 (0.5)
Weight (g)	1283 (273)	2049 (317)	3172 (394)
Parenteral hydration (mL/kg/day)	96 (19)	81 (18)	NA
Heart rate (bpm)	138 (19)	130 (14)	130 (15)
Systolic blood pressure (mmHg)	69 (12)	70 (8)	79 (5)
Mean blood pressure (mmHg)	48 (9)	50 (4)	49 (7)
Diastolic blood pressure (mmHg)	38 (9)	39 (4)	39 (4)
Non-invasive respiratory support	19 (63.3%)	0 (0%)	0 (0%)
No respiratory support	11 (36.7%)	30 (100%)	30 (100%)

Values are presented as mean (SD), median (range) and count (%). Abbreviations: GA = gestational age, NA = not applicable.

**Table 4 diagnostics-12-00350-t004:** Intraobserver and interobserver reproducibility of LAS measurements.

	Intraobserver Reproducibility		Interobserver Reproducibility	
	Bias (95% LOA)	ICC (95% CI)	Bias (95% LOA)	ICC (95% CI)
LASr	−0.4 (−1.5 to 2.4)	0.99 (0.967 to 0.998)	0.6 (−3.3 to 4.5)	0.936 (0.743 to 0.985)
LAScd	0.0 (−1.4 to 1.3)	0.993 (0.974 to 0.998)	−0.3 (−3.3 to 3.9)	0.938 (0.752 to 0.936)
LASct	−0.7 (−2.9 to1.4)	0.986 (0.944 to 0.996)	−0.6 (−6.1 to 4.8)	0.871 (0.532 to 0.969)

All ICC *p* values < 0.01. Abbreviations: LAS = left atrial strain; LASr = left atrial strain reservoir; LAScd = left atrial strain conduit; LASct = left atrial strain contraction; CI = confidence intervals; ICC = intraclass correlation coefficient; LOA = limits of agreement.

**Table 5 diagnostics-12-00350-t005:** LAS reference ranges at 24 and 48 h in the second cohort of neonates.

	Timepoint 1 (24 ± 2 h)	Timepoint 2 (48 ± 2 h)	*p*
**Neonates with GA ≤ 32 Weeks**			
LASr (%)	32.3 ± 5	36.8 ± 5.6	*
LAScd (%)	−18.6 ± 21.4	−20.0 ± 18.8	
LASct (%)	−11.4 ± 9.8	−13.7 ± 17.4	
**Neonates with GA between 32 and 37 Weeks**			
LASr (%)	33.3 ± 8.4	36.9 ± 13.4	*
LAScd (%)	−22.5 ± 11.4	−21.2 ± 24.6	
LASct (%)	−10.7 ± 10	−13.3 ± 8.4	*
**Neonates with GA ≥ 37 Weeks**			
LASr (%)	33.3 ± 4.4	36.7 ± 4.8	*
LAScd (%)	−20.6 ± 11.4	−21.6 ± 12.2	
LASct (%)	−13.4 ± 9.4	−13.4 ± 10.4	

Values are presented as mean ± 2SD. Abbreviations: GA = gestational age, LASr = left atrial strain reservoir; LAScd = left atrial strain conduit; LASct = left atrial strain contraction. * *p* < 0.05.

**Table 6 diagnostics-12-00350-t006:** Echocardiographic parameters at 24 and 48 h in the second cohort of neonates.

	Timepoint 1 (24 ± 2 h)	Timepoint 2 (48 ± 2 h)	*p*
**Neonates with GA ≤ 32 Weeks**			
SF (%)	31.1 (6.0)	32.8 (9.9)	
EF (%)	54.9 (7.8)	57.0 (7.2)	
E (cm/s)	43.4 (10.7)	40.8 (10.5)	
A (cm/s)	53.4 (14.0)	47.7 (7.6)	*
E/A	0.9 (0.2)	0.9 (0.2)	
e’ (cm/s)	5.6 (1.6)	6.0 (1.1)	
a’ (cm/s)	6.6 (2.7)	6.1 (2.0)	
E/e’	9.6 (3.4)	7.5 (2.1)	*
LVO (mL/kg/min)	178 (64)	195 (86)	
GLS (%)	−19.8 (3.9)	−19.7 (4.0)	
SR E (1/s)	1.7 (0.4)	1.9 (0.5)	
SR A (1/s)	2.1 (0.6)	2.0 (0.3)	
SR S (1/s)	−1.7 (0.2)	−1.7 (0.2)	
**Neonates with GA between 32 and 37 Weeks**			
SF (%)	32.4 (6.8)	30.2 (10.0)	
EF (%)	57.9 (11.7)	56.4 (8.9)	
E (cm/s)	48.8 (8.4)	46.5 (9.0)	
A (cm/s)	51.2 (10.7)	46.8 (8.4)	
E/A	0.96 (0.2)	0.96 (0.2)	
e’ (cm/s)	6.1 (2.0)	7.4 (4.9)	
a’ (cm/s)	6.4 (2.4)	6.5 (2.2)	
E/e’	8.5 (3.2)	7.4 (3.1)	
LVO (mL/kg/min)	151 (44)	152 (66)	
GLS (%)	−19.5 (4.1)	−18.6 (8.6)	
SR E (1/s)	2.0 (0.5)	2.2 (0.4)	
SR A (1/s)	1.8 (0.4)	1.9 (0.4)	
SR S (1/s)	−1.6 (0.3)	−1.6 (0.3)	
**Neonates with GA ≥ 37 Weeks**			
SF (%)	32.3 (4.9)	33.6 (6.9)	
EF (%)	57.9 (4.6)	56.5 (3.4)	
E (cm/s)	50.1 (11.9)	49.9 (8.1)	
A (cm/s)	52.3 (11.3)	50.9 (10.9)	
E/A	1.0 (0.2)	1.0 (0.2)	
e’ (cm/s)	6.5 (1.1)	6.0 (1.2)	
a’ (cm/s)	6.0 (1.7)	6.2 (2.0)	
E/e’	8.2 (2.0)	8.7 (1.4)	
LVO (mL/kg/min)	150 (27)	168 (29)	
GLS (%)	−21.0 (2.7)	−22 (3.2)	
SR E (1/s)	1.9 (0.4)	1.9 (0.4)	
SR A (1/s)	1.8 (0.4)	1.7 (0.4)	
SR S (1/s)	−1.6 (0.3)	−1.6 (0.3)	

Values are presented as mean (SD). Abbreviations: GA = gestational age, GLS = global longitudinal strain; SR = strain rate; SF = shortening fraction; EF = ejection fraction; LVO = left ventricular output. * *p* < 0.05.

**Table 7 diagnostics-12-00350-t007:** Correlations between LASr, LAScd, LASct and other echocardiographic parameters at 24 and 48 h in the second cohort of neonates.

	GLS	E/e’
**Timepoint 1 (24 ± 2 h)**		
LASr	−0.270 *	−0.270 *
LAScd	−0.162	+0.290 *
LASct	+0.05	+0.117
**Timepoint 2 (48 ± 2 h)**		
LASr	−0.347 *	−0.351 *
LAScd	+0.311	−0.118
LASct	+0.230	−0.027

LASr = left atrial strain reservoir; LAScd = left atrial strain conduit; LASct = left atrial strain contraction GLS = global longitudinal strain; SR = strain rate; SF = shortening fraction; EF = ejection fraction; LVO = left ventricular output. Pearson’s correlation coefficients are presented, with * at <0.05 significance level.

**Table 8 diagnostics-12-00350-t008:** Comparison of echocardiographic parameters in preterm neonates with and without hsPDA.

	Neonates with hsPDA	Neonates without hsPDA	*p*
E (cm/s)	48 (9.1)	41 (11.5)	
A (cm/s)	58.3 (10.2)	49.8 (15.5)	
E/A	0.87 (0.2)	0.93 (0.2)	
E/e’	8.9 (1.8)	9.7 (4.2)	
GLS (%)	−19 (1.7)	−18.7 (4.3)	
LASr (%)	30.4 (2.2)	33.3 (2.2)	*
LAScd (%)	−15.6 (4.1)	−16.2 (4.4)	
LASct (%)	−8.2 (3.8)	−13.1 (4.6)	*

Values are presented as mean (SD). Abbreviations: hsPDA = hemodynamically significant patent ductus arteriosus; GLS = global longitudinal strain; LASr = left atrial strain reservoir; LAScd = left atrial strain conduit; LASct = left atrial strain contraction. * *p* < 0.05.

## Data Availability

Data are available upon reasonable request.

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
