# Peer review of "Feasibility, Reproducibility and Reference Ranges of Left Atrial Strain in Preterm and Term Neonates in the First 48 h of Life"

_diagnostics, 2022, doi:10.3390/diagnostics12020350_

Round 1

Reviewer 1 Report

The manuscript by Ficial B. et al has presented the Left atrial strain analysis in neonates via echocardiography. The manuscript is well written and supported by its data. I have following comments for the manuscript:

  1. Please provide representative images of 4 chamber view at 24 hr and 48 hr time points and include them in main text.
  2. Since these kind of observations has previously done by others, please state why this analysis is better than others. Also provide supportive data to strengthen it.

Author Response

Point 1: The manuscript by Ficial B. et al has presented the Left atrial strain analysis in neonates via echocardiography. The manuscript is well written and supported by its data. I have following comments for the manuscript.

Please provide representative images of 4 chamber view at 24 hr and 48 hr time points and include them in main text.

Response 1: We thank the reviewer for their positive comment. Following the reviewer’s suggestions, we added a detailed description of our LAS protocol in the Appendix. In the Figure A1 we provided a representative image of a 4-chamber view ar 24 hour. We did not provide an image of a 4-chamber view at 48 hours due to space limitations and because it is pretty similar to the one at 24 hours. (Lines 828-855)

Point 2: Since these kind of observations has previously done by others, please state why this analysis is better than others. Also provide supportive data to strengthen it.

Response 2: We thank the reviewer for this observation. To the best of our knowledge, this is the first study that longitudinally assessed LAS measured by 2D STE in both term and preterm neonates in the first 48 hours of life. Previous studies presented data on LAS in term neonates beyond the first days of life. The only study in preterm neonates again reported data on LAS in outside the transtional period at 3 days and 4 weeks of life [DOI: 10.1111/echo.14140]. We have added discussion of such factors (Lines 539-554).

Reviewer 2 Report

Title: Feasibility, reproducibility and clinical significance of left atrial strain in preterm and term neonates in the first 48 hours of life.

Manuscript: diagnostics-1512358

Overview: This manuscript proposes to assess feasibility of left atrial strain (LAS) assessment in preterm and term neonates and provide normal values of LAS measurements in preterm and term neonates in the first 48 hours of life, assess LAS reproducibility and provide echocardiographic parameters.

Specific Points:

  1. The proposed premise of developing this technique is to be able to monitor diastolic dysfunction in neonates to provide both a reproducible systematic procedure that could lead to standardized clinical techniques to assess diastolic dysfunction, whether it be a normal developmental process or a pathological circumstance, and to provide a foundational trove of data to be used for future reference. I would encourage the authors to develop these points more robustly in the discussion and expand on the potential clinical benefits of this procedure.
  2. Line 190 “ Both LAScd and LASct did not show statistically significant differences between 24 and 48 hours in all three groups” The data in table 4 indicates that LASct is indeed significantly changes at the 48 hour timepoint as denoted by an asterisk. Also, please define asterisk symbol in the legend.
  3. A few grammatical issues. Line 231 compered agreement should read compared data. Line 239-240 “slightly higher feasibility and a lower time wasting.”should be changed to” superior feasibility and is less time consuming” as written in the limitations section. Line 241 change parameters to

Assessment: A well conducted study introducing a standardized technique to a novel assessment. The data is robust and the methods are performed with the necessary vigor. Pending minor revision I am in favor of accepting this manuscript.

Author Response

Point 1: Overview: This manuscript proposes to assess feasibility of left atrial strain (LAS) assessment in preterm and term neonates and provide normal values of LAS measurements in preterm and term neonates in the first 48 hours of life, assess LAS reproducibility and provide echocardiographic parameters.

Response 1: We thank the reviewer for putting our work into context.

Point 2: The proposed premise of developing this technique is to be able to monitor diastolic dysfunction in neonates to provide both a reproducible systematic procedure that could lead to standardized clinical techniques to assess diastolic dysfunction, whether it be a normal developmental process or a pathological circumstance, and to provide a foundational trove of data to be used for future reference. I would encourage the authors to develop these points more robustly in the discussion and expand on the potential clinical benefits of this procedure.

Response 2: We thank the reviewer for their positive comment. Following reviewer’s suggestions we added discussion on such factors (Lines 539-545, 604-729).

Point 3: Line 190 “ Both LAScd and LASct did not show statistically significant differences between 24 and 48 hours in all three groups” The data in table 4 indicates that LASct is indeed significantly changes at the 48 hour timepoint as denoted by an asterisk. Also, please define asterisk symbol in the legend.

Response 3: Thank you for pointing this out, we clarified that when comparing LASct at 24 and 48 hours in all 90 neonates we did not find a statistically significant differences. However, only in neonates with GA between 32 and 37 weeks we found a significat difference. We amended the text and defined asterix in the Table legend. (Line 324-346).

Point 4: A few grammatical issues. Line 231 compered agreement should read compared data.

Response 4: Thanks for this observation, we modified the sentence. (Lines 578-579)

Point 5: Line 239-240 “slightly higher feasibility and a lower time wasting.”should be changed to” superior feasibility and is less time consuming” as written in the limitations section.

Response 5: Thanks for this observation, we modified the sentence as suggested. (Lines 735-737)

Point 6: Line 241 change parameters to parameter.

Response 6: Thanks for this observation, we amended the text (Line 737-738).

Point 7: Assessment: A well conducted study introducing a standardized technique to a novel assessment. The data is robust and the methods are performed with the necessary vigor. Pending minor revision I am in favor of accepting this manuscript.

Response 7: We thank the reviewer for their positive comment.

Reviewer 3 Report

Very well written paper on diastolic function evaluation by LA strain. Although, there is no novelty of this parameter, it is feasibility study in neonates and pre-term infants are worth commending. I have few comments:

  1. In the absence of any case of ventricular dysfunction, it is hard to comment how the LA strain parameters will hold between observers and centers.
  2. If LA strain especially LAS has a good correlation with E/E', what is the need for adding another parameter to jumble up so many things for clinical use.
  3. Table-1: Add Heart Rate?
  4. What is the reason E/E' had good differentiation in preterm <32 weeks where as none in other 2 groups (Table-4)
  5. Any comment on LA strain in CHD patients especially with SV physiology
  6. Although it is feasible; there is still no agreement whether single plane or biplanar estimation of LA strain is useful?

Author Response

Point 1: Very well written paper on diastolic function evaluation by LA strain. Although, there is no novelty of this parameter, it is feasibility study in neonates and pre-term infants are worth commending. I have few comments:

Response 1: We thank the reviewer for their positive comment.

Point 2: In the absence of any case of ventricular dysfunction, it is hard to comment how the LA strain parameters will hold between observers and centers.

Response 2: We thank the reviewer for raising this important point. We aknowedged it in the discussion section (Lines 616-620). In the revised version of the manuscript we added a comparison between neonates with and without hsPDA (Lines 513-536 ). Although echocardiographic parameters of systolic and diastolic function did not show statistically signifcant differences between cases and controls, we found that LAS measurements can play an important role to early identify neonates with volume overload. We added discussion on such factors (Lines 621-729).

Point 3: If LA strain especially LAS has a good correlation with E/E', what is the need for adding another parameter to jumble up so many things for clinical use.

Response 3: We thank the reviewer for this observation. We feel that LAS is a better tool to assess LV diastolic function than Doppler and TDI parameters. It is highly feasible, reproducible, quick to use and it is not angle-dipendent. However, further studies are needed before a widespread application of LAS in clinical practice. In the revised version of the manuscript, we added data on neonates with hsPDA. In this category of patients, we showed that LAS, compared to other traditional parameters of diastolic function, may play a role in the early identification of neonates with volume overload. (Lines 513-536, 621-729).

Point 4: Table-1: Add Heart Rate?

Response 4: We appreciate the reviewer’s suggestions and we added heart rate (Lines 347-350 and Table 3).

Point 5: What is the reason E/E' had good differentiation in preterm <32 weeks where as none in other 2 groups (Table-4)

Response 5: Previous data showed reduction of E/e’ in very preterm neonates in the transitional period [. In this population dramatic changes occur in the first 48 hours of life leading to an increased pulmonary blood flow and increased loading conditions of LV. The latter may be exacerbated by the presence of a PDA, that is more common in neonates below 32 weeks of GA. Myocardial velocities measured by tissue Doppler are influenced by loading conditions and this may explain the decrease in E/e’ [http://dx.doi.org/10.1016/j.earlhumdev.2014.09.004]. We feel that this detailed explanation is beyond the scope of our paper, whose primary aim is to establish reference ranges of LAS by 2D STE that can be used as reference data for future research on diastolic dysfunction in neonates. However, we are happy to add discussion on such factors on the revised version of the manuscript, following the Editor’s and Reviewer’s judgment.

Point 6: Any comment on LA strain in CHD patients especially with SV physiology

Response 6: We thank the reviewer for this observations. In patients with functionally single ventricle heart disease, the single atrium is dilated and has reduced reservoir function, with increased reliance on atrial contraction for ventricular filling. LAS in single ventricle patients deviate significantly from normal controls and appear similar to patients with early LV diastolic dysfunction. We added discussion on such factors (Lines 607-611).

Although it is feasible; there is still no agreement whether single plane or biplanar estimation of LA strain is useful?

Response 6: We thank the reviewer for raising this important point. From a pratical perspective, we feel that monoplane LAS should be preferred, given the superior feasibility and the excellent agreement between monoplane and biplane LAS computatio. We added discussion on such factors (Lines 570-586).

Reviewer 4 Report

The authors of the paper report on the feasibility and the reproducibility of measuring left atrial strain (LAS) in preterm and term neonates, and provide reference ranges for LAS at 24 and 48 hours of life. Their primary goal was to study the feasibility of measuring LAS by developing a protocol for image acquisition and analysis, which they then applied to 30 preterm and term neonates. Their secondary goals were to evaluate the intra- and interobserver reproducibility of LAS measurements, to explore correlations with echocardiographic parameters of left ventricular diastolic and systolic function, and to provide reference ranges for LAS at 24 and 48 hours of life by studying a sample of 90 healthy preterm and term neonates. They conclude that measuring LAS is highly feasible and reproducible in preterm and term neonates in the first 48 hours of life.

The premise of the paper is interesting and important, as there is very little published data on LAS in neonates, and it provides a useful contribution to the growing literature on LAS in children. The methodology used seems appropriate and the data support the author’s conclusions.

Major comments:

  1. Consider revising the title as “clinical significance of LAS” was not explored in the study. This is even explicitly stated by the authors in the limitations section (line 278).
  2. The sample size is too small to provide normal values. It would be more accurate to use the term “reference ranges”, which should be presented as mean ± 2 standard deviations.
  3. How many patients had a PDA? Was there any evidence of left atrial volume load in these patients? Since LAS can be affected by a significant PDA, this should be reported by the authors.
  4. The authors mention in the methods section a series of echocardiographic parameters that were obtained in order to explore their correlation with LAS. However, correlation with many of these parameters is not presented in the results section. Why?
  5. Heart rate is not reported for any of the measurements, while it is known that all components of atrial function are dependant on HR. Reporting the HR would therefore add to the robustness of the study.
  6. Figure 2 seems superfluous. Feasibility is a simple calculation and reporting it in the text only is sufficient.
  7. Data would be more clearly presented if Table 4 were divided into 2 tables, one presenting only the LAS measurements and the other presenting the correlations between LAS measurements and the other echo parameters. This would also be much more in accordance with how the data is presented in the text.
  8. Line 190: the authors state there was no significant difference in LASct at 24 vs. 48 hours, but then indicate a significant difference for the 32-37 weekers in Table 4. Please rectify.
  9. The manuscript could greatly benefit from a better structured pattern for the Discussion. The authors could first describe only what their data shows, then explain the significance of their results, and finally provide the critical analysis of their data. The first 3 paragraphs seem to reiterate points from the introduction, and the points discussed in Lines 237-241 and 265-269 also seem to be discussed in the limitations section.

Minor comments:

  1. The authors state in the abstract and on Line 213 that LAS/2D STE is the gold standard for evaluation of LV filling pressures. Is this really established as a gold standard?
  2. A couple of additional references are suggested to show the existing literature on LAS in children https://doi.org/10.1016/j.echo.2018.10.002, https://doi.org/10.1016/j.echo.2017.10.011
  3. Lines 35, 91 and 102: references seem to be missing
  4. Line 139: did the authors mean “in random order” instead of “in randomly chosen subjects”?
  5. Lines 201-203: it seems unclear what the r values represent. If it’s the Pearson correlation coefficients, they seem to be too close to 0 to be statistically significant. Please clarify.

Round 2

Reviewer 4 Report

Dear authors,

Thank you for addressing the comments from the first review. Your research is of significant interest and therefore worth publishing. However, further improvements are suggested prior to recommending acceptance:

  1. Do not repeat in the manuscript data that is clearly presented in Tables 2, 4, 6 and 7. For example, on lines 183-185 “Comparison between monoplane LAS vs biplane LAS showed an excellent agreement, with intraclass correlation coefficients 0.967, 0.959, 0.937 for LASr, LAScd and LASct respectively (Table 2).” could be replaced by “There was excellent agreement and correlation between monoplane and biplane LAS measurements (Table 2).”
  1. Reorder the results section as follows:
    1. Lines 230-233 would better follow after line 196, but without repeating the values (since already presented in the table), and providing the value
    2. Reliability data would better follow after the feasibility data on line 201. Table 6 should then become Table 4, with tables 4 and 5 becoming tables 5 and 6. Again, for each table, there is no need to repeat the values in the manuscript, simply state what the data shows.
  1. Present correlations between LAS and functional measurements in a table. There should be enough space after removing all the text that is repeated from the other tables. Then describe the correlations in the manuscript, for example “there was a weak negative linear correlation between LASr and both GLS and E/e’.” (Apologies for not being clearer in the first review).
  2. Although much improved, the discussion would flow better if the order followed the results section. After the first paragraph, start with discussing the protocol and how it allowed to demonstrate great feasibility and reliability of the measurements (lines 290-320 + lines 367-372). Then, discuss how this allowed to provide reference ranges (lines 281-289) and explain their significance (lines 374-382). Finally, discuss the significance of the correlation with functional measurements and the impact of hsPDA (fine as it is).

Author Response

Response to Reviewer 4 Comments

Point 1: Dear authors,

Thank you for addressing the comments from the first review. Your research is of significant interest and therefore worth publishing. However, further improvements are suggested prior to recommending acceptance:

Response 1: We thank the reviewer for their positive comment.

Point 2: Do not repeat in the manuscript data that is clearly presented in Tables 2, 4, 6 and 7. For example, on lines 183-185 “Comparison between monoplane LAS vs biplane LAS showed an excellent agreement, with intraclass correlation coefficients 0.967, 0.959, 0.937 for LASr, LAScd and LASct respectively (Table 2).” could be replaced by “There was excellent agreement and correlation between monoplane and biplane LAS measurements (Table 2).”

Response 2: We modified the text following the reviewer suggestions.

Point 3: Reorder the results section as follows:

Lines 230-233 would better follow after line 196, but without repeating the values (since already presented in the table), and providing the p value.

Reliability data would better follow after the feasibility data on line 201. Table 6 should then become Table 4, with tables 4 and 5 becoming tables 5 and 6.

Response 3: We modified the text following the reviewer suggestions.

Point 4: Again, for each table, there is no need to repeat the values in the manuscript, simply state what the data shows.

Response 4: We modified the text following the reviewer suggestions.

Point 5: Present correlations between LAS and functional measurements in a table. There should be enough space after removing all the text that is repeated from the other tables. Then describe the correlations in the manuscript, for example “there was a weak negative linear correlation between LASr and both GLS and E/e’.” (Apologies for not being clearer in the first review).

Response 5: We added the table and we modified the text following the reviewer suggestions.

Point 6: Although much improved, the discussion would flow better if the order followed the results section. After the first paragraph, start with discussing the protocol and how it allowed to demonstrate great feasibility and reliability of the measurements (lines 290-320 + lines 367-372). Then, discuss how this allowed to provide reference ranges (lines 281-289) and explain their significance (lines 374-382). Finally, discuss the significance of the correlation with functional measurements and the impact of hsPDA (fine as it is).

Response 6: We appreciate this observation and we modified the text following the reviewer suggestions.